# Defining substrate requirements for cleavage of farnesylated prelamin A by the integral membrane zinc metalloprotease ZMPSTE24

**Kaitlin M. Wood, Eric D. Spear, Otto W. Mossberg, Kamsi O. Odinammadu, Wenxin Xu¤, Susan Michaelis**\*

Department of Cell Biology, The Johns Hopkins University School of Medicine, Baltimore, Maryland, United States of America

¤ Current address: Dept. of Biology, Southern University of Science and Technology, Shenzhen, China
\* michaelis@jhmi.edu

## Abstract

The integral membrane zinc metalloprotease ZMPSTE24 plays a key role in the proteolytic processing of farnesylated prelamin A, the precursor of the nuclear scaffold protein lamin A. Failure of this processing step results in the accumulation of permanently farnesylated forms of prelamin A which cause the premature aging disease Hutchinson-Gilford Progeria Syndrome (HGPS), as well as related progeroid disorders, and may also play a role in physiological aging. ZMPSTE24 is an intriguing and unusual protease because its active site is located inside of a closed intramembrane chamber formed by seven transmembrane spans with side portals in the chamber permitting substrate entry. The specific features of prelamin A that make it the sole known substrate for ZMPSTE24 in mammalian cells are not well-defined. At the outset of this work it was known that farnesylation is essential for prelamin A cleavage *in vivo* and that the C-terminal region of prelamin A (41 amino acids) is sufficient for recognition and processing. Here we investigated additional features of prelamin A that are required for cleavage by ZMPSTE24 using a well-established humanized yeast system. We analyzed the 14-residue C-terminal region of prelamin A that lies between the ZMPSTE24 cleavage site and the farnesylated cysteine, as well 23-residue region N-terminal to the cleavage site, by generating a series of alanine substitutions, alanine additions, and deletions in prelamin A. Surprisingly, we found that there is considerable flexibility in specific requirements for the length and composition of these regions. We discuss how this flexibility can be reconciled with ZMPSTE24's selectivity for prelamin A.

## Introduction

Membrane associated proteases are important in a variety of physiological and medically relevant processes through their cleavage of specific cellular substrates [1, 2]. Defining the critical features for substrate recognition is key to understanding how specific proteases function and for developing therapeutic strategies for diseases in which they are involved. Our studies are focused on ZMPSTE24, an integral membrane zinc metalloprotease important for human health and longevity [3–5]. ZMPSTE24 mediates a critical proteolytic processing event in the

---

**Data Availability Statement:** All relevant data are within the manuscript and its Supporting information files.

**Funding:** This work was funded by a National Institute of Health (NIH) grant to SM (5R35GM127073). Kaitlin Wood was funded in part by a Hay Fellowship from the Department of Cell Biology, Johns Hopkins School of Medicine. The funders had no role in study design, data collection and analysis, decision to publish, or preparation of the manuscript. The funder provided support in the form of salaries for author [KMW], but did not have any additional role in the study design, data collection and analysis, decision to publish, or preparation of the manuscript. The specific roles of these authors are articulated in the 'author contributions' section.

**Competing interests:** The authors have declared that no competing interests exist.

maturation of the farnesylated protein prelamin A, the precursor of the nuclear scaffold component lamin A encoded by *LMNA* [5–7]. Mutations in *LMNA* or *ZMPSTE24* that prevent prelamin A cleavage cause the severe premature aging disorder Hutchinson-Gilford Progeria syndrome (HGPS) or the related progeroid diseases, mandibuloacral dysplasia type-B (MAD-B) and restrictive dermopathy (RD) [8–15]. Normal maturation by ZMPSTE24 cleaves off the C-terminal 15 residues of prelamin A, including its farnesyl-cysteine (Fig 1A and 1B). In progeroid diseases an aberrant, uncleaved, and permanently farnesylated form of lamin A is the "molecular culprit" that causes disease symptoms [16, 17]. Importantly, diminished prelamin A processing by ZMPSTE24 may also be a critical factor in normal physiological aging [18].

The X-ray crystallography structures of human ZMPSTE24 and its yeast counterpart, Ste24, have revealed novel features that define a unique class of membrane protease [5, 19–21]. The seven transmembrane spans of ZMPSTE24/Ste24 form a voluminous water-filled intramembrane barrel-shaped chamber that is capped at both ends, large enough to fit a 10kD protein (Fig 1C). Notably, the zinc metalloprotease catalytic motif (HEXXH. . ...E; shared among the gluzincin subfamily of metalloproteases [22, 23]), is located inside of the ZMPSTE24 chamber interior and thereby substrate access to the active site is highly restricted. The C-terminal portion of farnesylated prelamin A, the sole known substrate of this enzyme in mammalian cells, must enter into the chamber interior through one of several side portals evident in the structure. Defining the properties needed for substrate recognition, portal entry, and positioning of substrate will help clarify our understanding of the mechanism of prelamin A cleavage by ZMPSTE24. We have recently shown through comprehensive mutagenesis of residues surrounding the cleavage site of prelamin A (TRSY^LLGN) that having two hydrophobic residues just C-terminal to the scissile bond in the P1' and P2' positions (the two leucines in wild-type prelamin A) is critical for its cleavage by ZMPSTE24 [24]. In some proteases a region distant from the active site, termed an exosite, can facilitate the capture and proper orientation of substrate for cleavage. Whether an exosite exists in ZMPSTE24 for prelamin A positioning has not been examined.

The nuclear lamins A, B, and C are homologous intermediate filament proteins. Lamins polymerize through their alpha-helical coiled-coil rod domain to form a mesh-like network of homopolymers, called the lamina that underlies the inner nuclear membrane [25]. The nuclear lamina regulates the structure and shape of the nucleus and provides an organizing scaffold for heterochromatin [25–28]. Lamins A and B, but not lamin C, have a C-terminal CAAX motif in which C is cysteine, A is often aliphatic, and X is any residue. This motif directs a series of post-translational modification events, including farnesylation of the CAAX motif cysteine, endoproteolytic removal of the terminal three (AAX) residues, and carboxylmethylation of the farnesyl-cysteine. As is the case for lamin B and essentially all other CAAX proteins, farnesylation contributes to membrane binding and is generally a permanent post-translational modification [29]. However, for prelamin A, cleavage by ZMPSTE24 removes the C-terminal 15 residues which includes the farnesylated cysteine (Fig 1A and 1B). The biological purpose for why prelamin A undergoes farnesylation only to subsequently cleave off the modification is unclear. However, this ZMPSTE24-mediated cleavage is critical for human health, and failure to cleave leads to premature aging diseases HGPS, MAD-B, and RD, in which the disease severity corresponds to the extent of the prelamin A processing defect and the amount of permanently farnesylated prelamin A that accumulates [9, 30].

Ste24, the budding yeast *Saccharomyces cerevisiae* homolog of ZMPSTE24, is the first identified member of this protease subfamily. Yeast does not encode nuclear lamins. Rather, Ste24 was discovered based on its role in the biogenesis of the yeast mating pheromone **a**-factor, whose precursor also terminates with a CAAX motif [31, 32]. The **a**-factor precursor undergoes stereotypical CAAX processing, including farnesylation, -AAX cleavage, and carboxyl methylation. CAAX processing of the **a**-factor precursor is followed by two subsequent

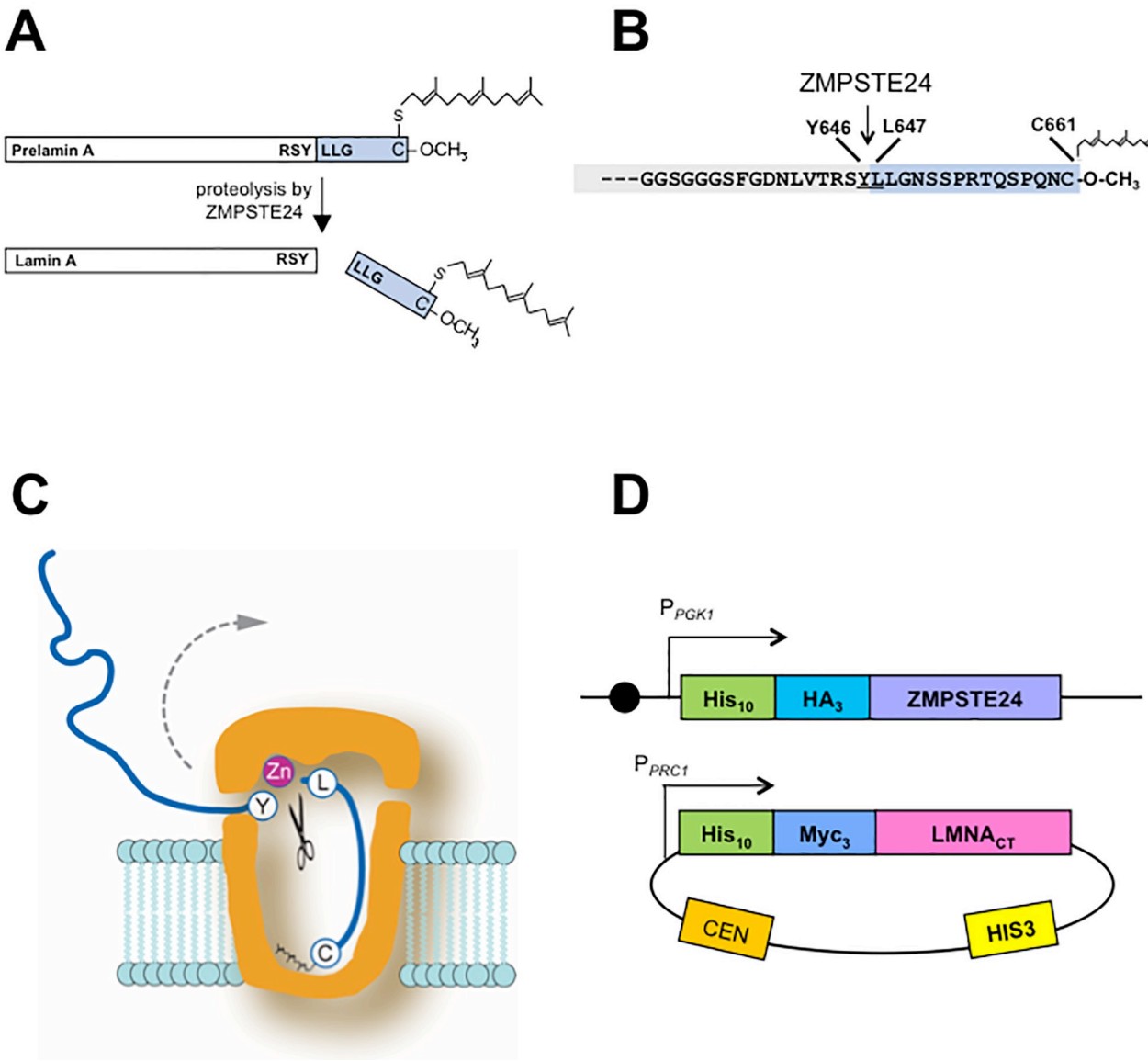

**Fig 1. Cleavage of human prelamin A by ZMPSTE24.** (A) The lamin A precursor prelamin A is 664 amino acids in length. CAAX processing of the C-terminal cysteine produces a farnesylated and carboxymethylated substrate that is cleaved by ZMPSTE24. The site of ZMPSTE24 cleavage that generates mature lamin A is indicated. (B) Detailed view of the C-terminal 34 residues of prelamin A showing that ZMPSTE24 cleavage occurs between Y646 and L647 of the farnesylated prelamin A precursor; the 15 amino acid segment that contains the farnesylated cysteine and is boxed in blue is cleaved off. (C) Schematic of the ZMPSTE24 protease chamber with prelamin A positioned for cleavage, showing the scissile bond and the farnesylated cysteine at the bottom of the chamber [5, 19]. The dotted arrow suggests how the substrate exits after cleavage. (D) The humanized yeast co-expression system with human ZMPSTE24 chromosomally integrated in strain SM6303 and the C-terminal portion of human *LMNA*-encoded prelamin A (His$_{10}$myc$_3$-LMNA$_{431-664}$) expressed from a yeast centromeric plasmid. The strains and plasmids are described in the Materials and Methods. Circle indicates a centromere.

cleavages, one of which is mediated by Ste24 [32–37]. It should be noted that both Ste24 and the intramembrane protease Rce1 can redundantly perform the -AAX cleavage for the **a**-factor precursor.[32, 35, 37].

The farnesylated **a**-factor precursor is the sole known specific substrate for Ste24 in yeast. However, genetic studies have suggested additional functions for Ste24 in ER-associated degradation (ERAD), the unfolded protein response (UPR), clearance of clogged translocons,

membrane protein topology, and secretion of proteins lacking signal sequences [38–43]. A role in protection against enveloped viruses has also been reported for ZMPSTE24 in mammalian cells [44, 45]. In these roles, it is unlikely that ZMPSTE24/Ste24 cleaves specific substrates, or that farnesyl is required for substrate processing. ZMPSTE24/Ste24 resides in both the endoplasmic reticulum (ER) membrane and the inner nuclear membrane (INM) in mammalian and yeast cells [46]. Thus, the cell may have evolved distinct pools of this protease that are dedicated for cleavage of various substrates.

Our goal in the present work is to gain a better understanding of the requirements for prelamin A as a ZMPSTE24 substrate. We have previously established that the C-terminal 41 amino acids of prelamin A contain all of the information needed for ZMPSTE24 processing *in vivo* [47]. Furthermore, our work and that of others has shown that farnesylation is critical for prelamin A cleavage by ZMPSTE24. Changing the cysteine in the prelamin CAAX motif to a serine or treating cells with a farnesyl transferase inhibitor (FTI) blocks farnesylation and also blocks ZMPSTE24 cleavage [48–50]. It has been proposed that farnesyl binds to the inside of the ZMPSTE24 on the chamber "floor" and helps to position prelamin A for cleavage, as shown in Fig 1C. [5, 19]. Here we have investigated the importance of the residues and length of the region between farnesyl and scissile bond for cleavage. We have also queried the role of the region N-terminal to the scissile bond for prelamin A cleavage *in vivo*. Our results indicate a surprising flexibility in the requirements for recognition of prelamin A for ZMPST24 and suggest alternative models for recruitment and positioning of the substrate relative to the protease chamber.

## Materials and methods

### Strains and plasmids used in this study

Strain SM6303 (*ste24::KanMX TRP1::NatMX-P_{PGK1}-His_{10}-HA_{3}-ZMPSTE24*) was used to measure cleavage efficiency of prelamin A variants, and has been previously described [24, 51]. It is derived from *ste24::KanMX met15Δ leu2Δ his3Δ ura3Δ MAT***a** (BY4741 Deletion Collection) and contains two integrated copies of yeast codon-optimized human *ZMPSTE24*.

Plasmids used here are derived from pSM3393 [24, 51], a *HIS3/CEN*-based plasmid that expresses the C-terminal residues 431–664 from human *LMNA*, and thus lacks the prelamin A coiled-coiled domain. The prelamin A coding sequence is expressed from the *PRC1* promoter and has an N-terminal $His_{10}myc_{3}$- epitope tag to detect proteins by western blotting. All prelamin A variants were constructed using mutagenic PCR and assembled using NEBuilder HiFi Assembly (New England Biolabs, Ipswich, MA).

To construct plasmid pSM3354, which contains the C-terminal 41 amino acids of LMNA fused to GFP, the coding sequence for GFP was amplified by PCR and inserted into pSM3393 to replace the $His_{10}$ tag. Variants of pSM3354 that remove or replace amino acids within the *LMNA* sequence were constructed using mutagenic PCR and NEBuilder® HiFi Assembly, as described above. After mutations were confirmed by Sanger sequencing, individual plasmids were transformed into SM6303 using standard lithium acetate methods, and selected on minimal medium lacking histidine. Plasmids used in this study are available upon request.

### Prelamin A cleavage assay

ZMPSTE24-mediated prelamin A cleavage was assessed essentially as described previously [24, 51]. Briefly, yeast strains were grown overnight in minimal synthetic complete dropout medium, back-diluted in fresh medium and grown for an additional 4–6 hours to an $OD_{600}$ of 1.0–1.2. Cells (1.5–2.0 $OD_{600}$ units) were collected by centrifugation in a microfuge and washed once with distilled water. Cells were pre-treated with 0.1M NaOH, re-pelleted, and lysed in SDS protein sample buffer at 65˚C for 10 minutes, followed by centrifugation at

21,000 x **g** for 2 minutes [52]. SDS-solubilized proteins (0.2–0.3 OD$_{600}$ cell equivalents) were resolved by 10% SDS-PAGE prior to transfer to nitrocellulose membranes using the Trans-Blot$^{®}$ Turbo™ system (Bio-Rad Laboratories, Hercules, CA) and blocked with Western Blocking Reagent (Roche, Indianapolis, IN). Lamin A proteins were detected with mouse anti-myc antibodies (clone 4A6, Millipore cat #05–724; 1:10,000 dilution) decorated with goat anti-mouse secondary IRDye 680RD antibodies (LI-COR, Lincoln, NE; 1:20,000 dilution). To confirm equal loading, hexokinase was detected with rabbit anti-hexokinase (1:200,000 dilution) decorated with goat anti-rabbit secondary IRDye 800CW antibodies (LI-COR; 1:200,000 dilution). Western blots were visualized using the Odyssey$^{®}$ imaging platform (LI-COR) and bands were quantified using Image Studio software (LI-COR). Relative percent processing of substrates was calculated by dividing the cleaved signal by the total signal (cleaved + uncleaved), and normalized to wild-type processing efficiency.

## Results

### The identity of most residues between the cleavage site and farnesylated cysteine is not critical for substrate proteolysis by ZMPSTE24

Prelamin A is proteolytically cleaved between residues Y646 and L647 by ZMPSTE24 to produce mature lamin A (Fig 1A and 1B) [6, 7, 53, 54]. We previously developed an *in vivo* humanized yeast assay that recapitulates ZMPSTE24-mediated processing of prelamin A in *Saccharomyces cerevisiae* [30, 51]. The yeast strain used in this assay lacks endogenous *STE24*, and instead expresses epitope-tagged human *ZMPSTE24* integrated in the genome, and a plasmid-borne form of prelamin A that contains amino acids 431–664 encoded by the *LMNA* gene (Fig 1D). This co-expression system typically yields ~85–90% mature, cleaved, lamin A [51]. As in mammalian cells, prelamin A cleavage in yeast requires farnesylation of the cysteine residue in the CAAX motif and an intact ZMPSTE24 cleavage site [24, 30]. Other than these requirements, little is known about sequence or structural features that make prelamin A the only known endogenous substrate of ZMPSTE24. Using this yeast assay system, we queried whether there are other features in the C-terminus of prelamin A needed for efficient cleavage by ZMPSTE24.

We first examined the region between the scissile bond and the farnesylated cysteine to determine whether there is information in this region critical for ZMPSTE24-dependent cleavage by performing step-wise alanine scanning mutagenesis (Fig 2A). For the wild-type prelamin A substrate we observed ~90% cleavage, which we then normalized to 100%, to represent the maximum cleavage (Fig 2B, lane 1 "WT"). Increasing the alanine substitutions upstream of the farnesylated cysteine did not dramatically affect cleavage until reaching S651 (10A), where cleavage fell to less than 50% (Fig 2B, lane 8). With additional alanine substitutions there was a precipitous drop in processing, with negligible mature lamin A being generated for the substitutions of 11A-14A (Fig 2B, lanes 10–13) which approach the scissile bond. Although it is conceivable that the decreased ZMPSTE24 cleavage seen for constructs ≥10A might reflect an indirect effect on farnesylation, this is unlikely, since the sole determinants of farnesylation efficiency are thought to be CAAX motif residues themselves [29, 55, 56]. These data indicate that the C-terminal portion of prelamin A harbors few or no specific residues that are absolutely necessary for ZMPSTE24 cleavage until the cleavage site itself is impaired.

### Determining the maximum and minimum length permitted between the ZMPSTE24 cleavage site and the farnesylated cysteine in prelamin A

A hydrophobic patch located near the "floor" of the ZMPSTE24 chamber has been proposed to bind the farnesyl moiety of prelamin A, helping to align its scissile bond with the catalytic

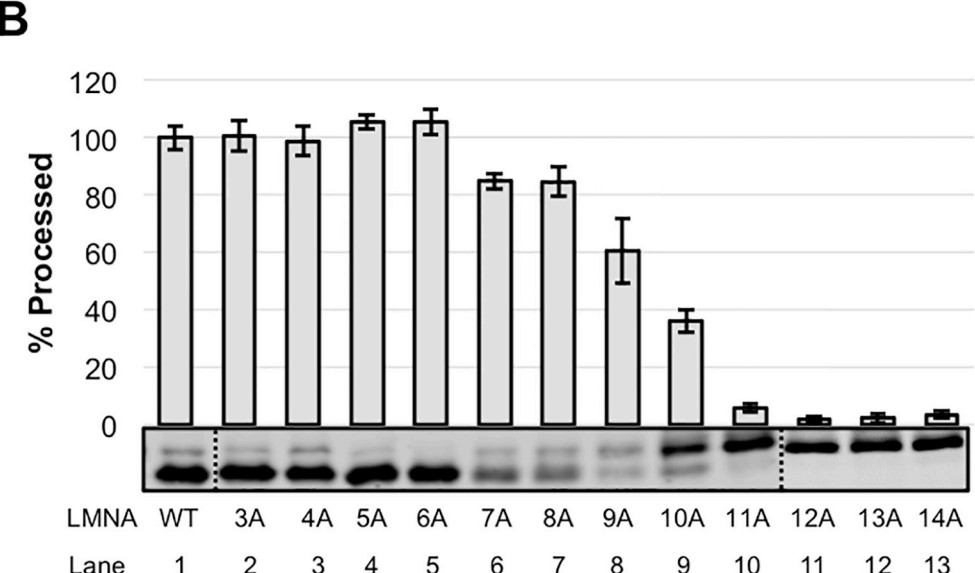

**Fig 2. Effect of C-terminal alanine substitutions in prelamin A on ZMPSTE24 cleavage.** (A) The C-terminal sequence of LMNA$_{643-664}$ is shown. The ZMPSTE24 cleavage site between Y646-L647 and the farnesyl on the cysteine of the CAAX motif are indicated. Alanine substitution mutations (red) are shown, with the number of residues changed denoted. (B) Strain SM6303 (*ste24Δ His$_{10}$HA$_3$-ZMPSTE24*) expressing WT or the indicated alanine substitution mutants were analyzed by SDS-PAGE and western blotting with anti-myc antibody. The percent of substrate processed relative to WT (set to 100%) was determined from 3 independent experiments, with the average and standard deviation of the mean shown as error bars. A representative gel is shown. Dotted line indicates where separate gels were spliced together.

residues at the opposite end of the ZMPSTE24 chamber ([19]; Fig 1D). We asked whether the spacing between the farnesylated cysteine and the scissile bond was important for substrate cleavage. First, we constructed a series of prelamin A substrates with increasing numbers of alanines in between N660 and C661, ranging from 1 to 19 extra residues (Fig 3A). The

insertion of up to 12 alanines preceding the farnesylated cysteine was well-tolerated, retaining almost 80% cleavage compared to WT (Fig 3B, lanes 1–7). Increasing the number of alanines further resulted in a step-wise decrease in processing, with 19 extra alanines dramatically diminishing cleavage to <5% (Fig 3B, lanes 9–13). Thus, ZMPSTE24 was able to accommodate a significant increase in length between the farnesylated cysteine and the cleavage site of prelamin A (from 14 up to 26 residues), and still maintain robust processing.

We next asked whether shortening the length between the farnesylated cysteine and cleavage site would impact cleavage efficiency. We generated a deletion series that started with deletion of residue P653, midway between the cleaved scissile bond and the farnesylated cysteine and extended in both directions (Fig 4A). We observed a gradual decrease, yet still saw proficient cleavage (>65%) for substrates with up to five residues deleted (Δ1-Δ5) (Fig 4B, lanes 1–6). Shortening the C-terminus further resulted in poor cleavage (Δ6 and Δ7; down to 42% and 18% of WT, respectively) (Fig 4B, lanes 7 and 8). These data are consistent with the results of the alanine substitution indicating that the C-terminal portion of prelamin A harbors few or no specific residues that are absolutely necessary for ZMPSTE24 cleavage. It should be noted that a diminished amount of total signal is evident for a number of the deletion constructs, a phenomenon we have occasionally also observed for other prelamin A mutants. The basis for this low expression is unknown and may be poor mRNA translatability or protein stability; however, low expression does not appear to significantly affect processing efficiency [24]. Overall, shortening the length between the farnesylated cysteine and the cleavage site from 14 to 7 residues diminishes, but does not fully ablate processing. The minimal processing observed in constructs shortened to 7 or 8 residues between the farnesylated cysteine and the cleavage site may reflect an altered positioning of the substrate within the ZMPSTE24 chamber.

## Analysis of the region N-terminal to the ZMPSTE24 cleavage site in prelamin A

We previously demonstrated in mammalian cells that the C-terminal 41 amino acids of prelamin A fused to GFP was proficient for ZMPSTE24-mediated cleavage, but that cleavage was significantly impeded when 29 or fewer amino acids were present in the GFP fusion construct [47]. This finding suggested either that specific residues that are present at the N-terminus of the 41-mer and absent in the 31-mer are important for recognition by ZMPSTE24 *in vivo*, or that the shortened 31-mer encountered a structural challenge due to its proximity to a folded domain, namely GFP. The sequence $G_{630}GSGGGS_{636}$ conforms to a classical "flexible linker sequence" (Fig 5A, gray box), and is present in the 41-mer but not the 31-mer. We considered that this specific sequence might potentially contribute to the observed cleavage differences.

To analyze this region in more detail we turned to our yeast assay. We constructed a GFP-41-mer plasmid (GFP-myc$_3$-LMNA$_{624-664}$; Fig 5A) and confirmed by western blotting that the GFP-41-mer was efficiently processed by ZMPSTE24 in the yeast system (Fig 5B, lane 1). Consistent with our previous study in HEK293 cells [47], shortening this region dramatically impaired cleavage, with the 31-mer showing < 50% cleavage and the 29-mer < 20% that of the 41-mer (Fig 5B, compare lanes 6 and 7 to lane 1). If the deleted region between R624-G634 contained specific amino acid information necessary for ZMPSTE24 cleavage (such as the flexible linker or other particular residues), then adding 10 alanines back to the 31-mer (31+10ala) would not be predicted to correct cleavage. However, if simply the length of the prelamin A C-terminus is critical for cleavage in the context of the GFP fusion, then lengthening the 31-mer back to 41 residues would suppress the cleavage defect, even though specific amino acids are absent. Indeed, extending the 31-mer with 10 alanines significantly enhanced cleavage from 40% to 69%, respectively (Fig 5C and 5D, compare lanes 2 and 3). Likewise, cleavage of the

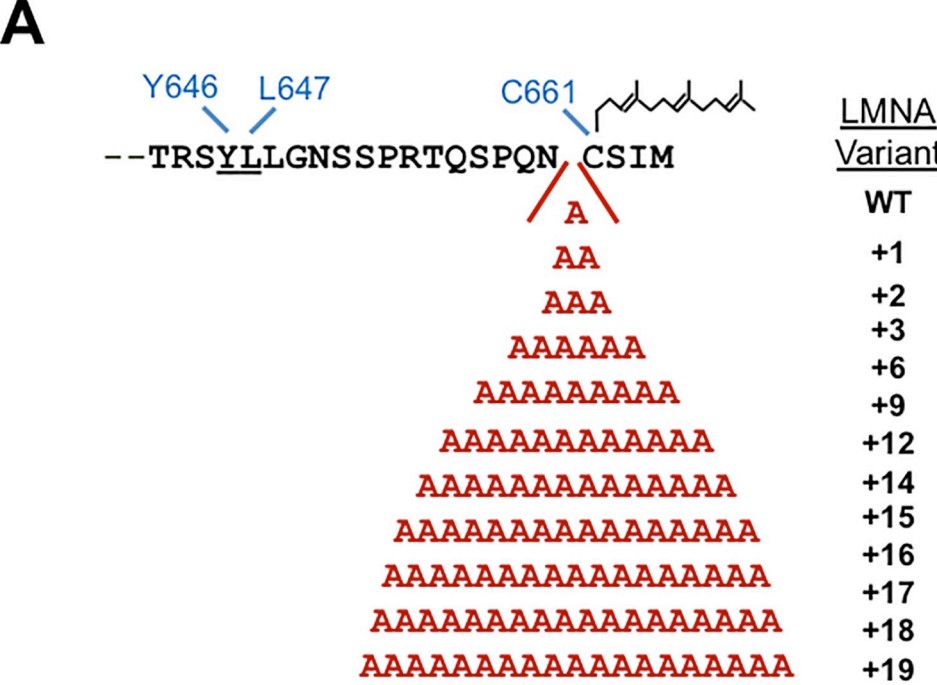

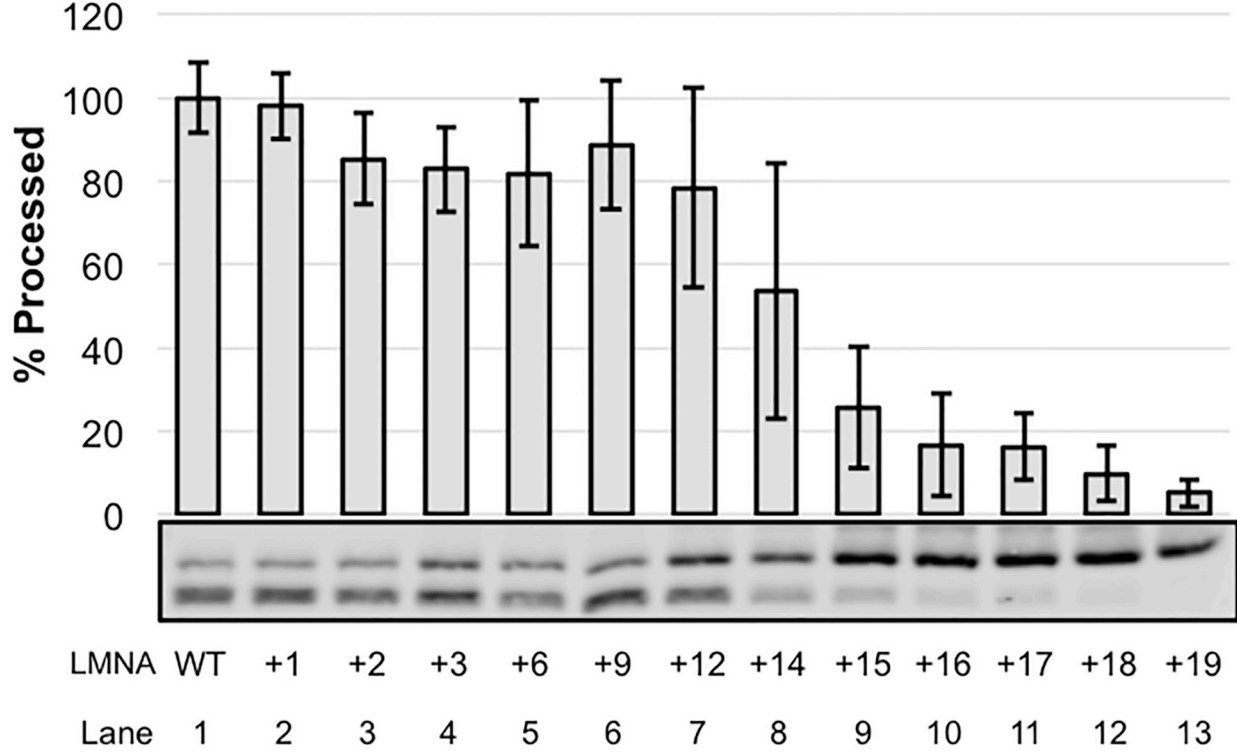

**Fig 3. Effect of C-terminal alanine insertions in prelamin A on ZMPSTE24 cleavage.** (A) Alanines (from 1 to 19 extra amino acids) were introduced just upstream of the CAAX cysteine (C661), in the prelamin A C-terminus (see Fig 1 for description of this region). (B) Alanine insertion variants were analyzed for cleavage by SDS-PAGE and western blotting as described in Fig 2.

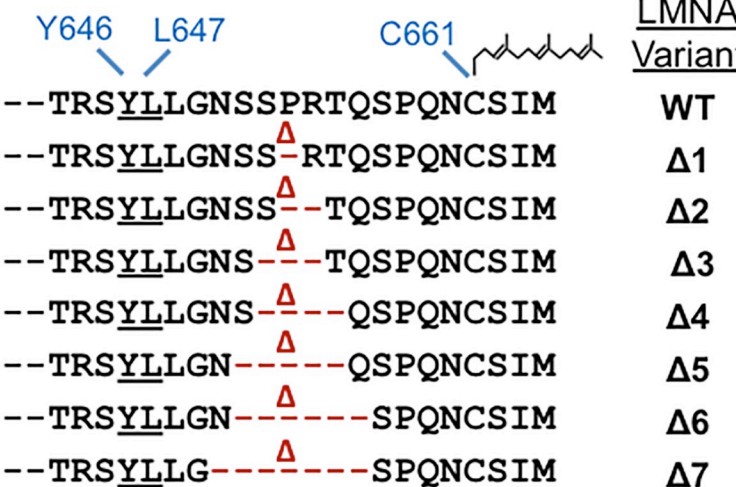

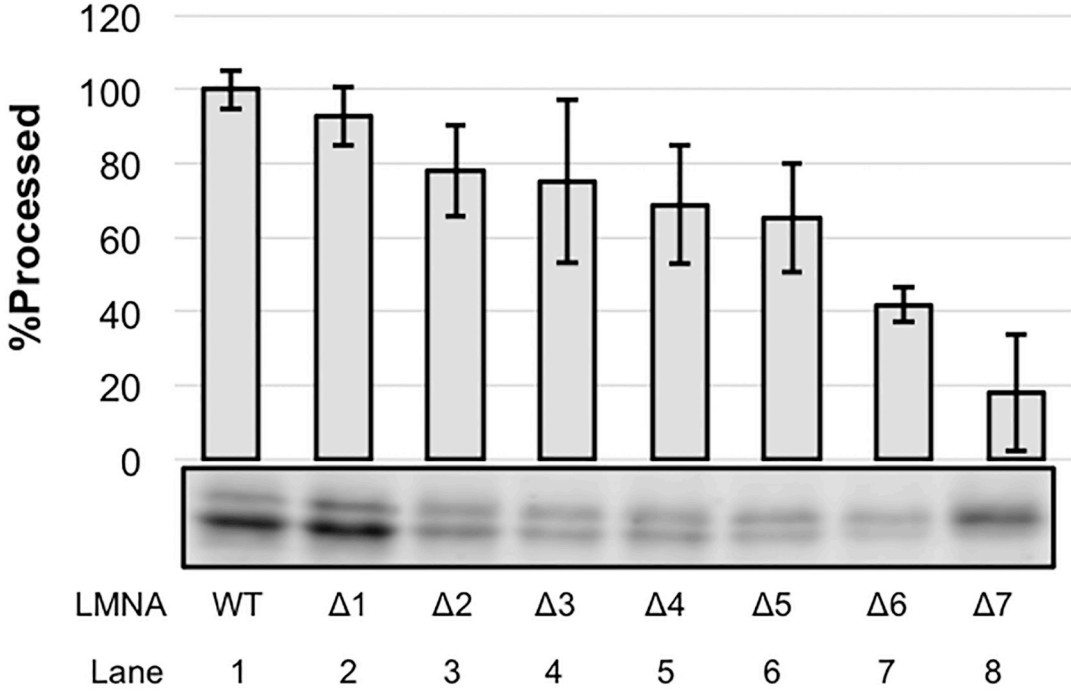

**Fig 4. Effect of shortening the prelamin A C-terminus on ZMPSTE24 cleavage.** (A) Prelamin A deletion mutations are indicated (B) The indicated deletion (Δ) mutants were analyzed for ZMPSTE24-mediated cleavage, as described in Fig 2.

29-mer and 27-mer was dramatically improved by extension with a string of alanine residues to bring their length back to 41 (Fig 5C and 5D, compare lanes 4 and 6 to lanes 6 and 7, respectively). These results show that sufficient length between the folded domain of GFP and the ZMPSTE24 cleavage site is important in the context of the GFP-41-mer, that a flexible linker is

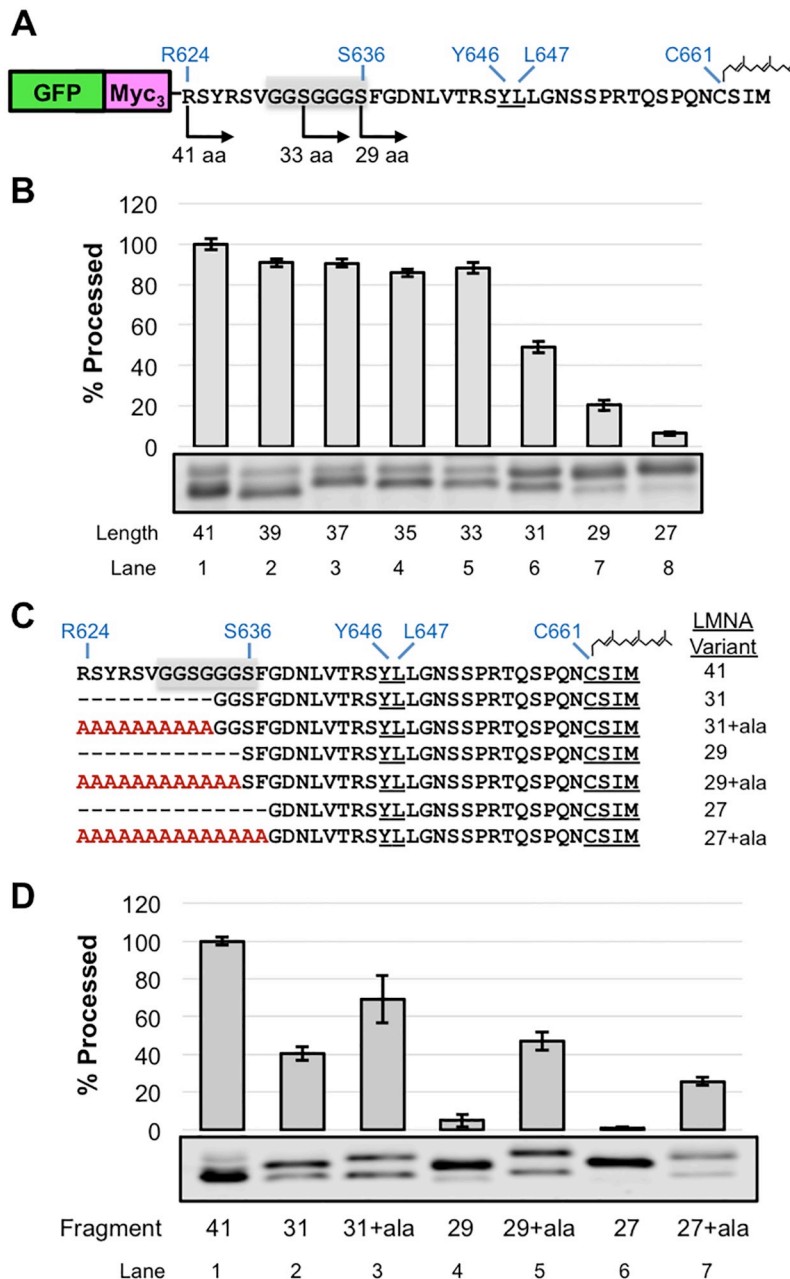

**Fig 5. Effect of changing the length and composition of the region N-terminal to the ZMPSTE24 cleavage site in the GFP-*LMNA*-41-mer "minimal substrate".** (A) Schematic of the GFP-myc$_3$-LMNA$_{624-664}$ substrate, that contains the C-terminal 41 residues of prelamin, referred to as the GFP-41-mer in the text. A subset of deletion endpoints tested are shown for orientation. The cleavage site residues (Y646-L647), farnesylated cysteine (C661), and a putative flexible linker (G$_{630}$-S$_{636}$), shaded in grey, are shown. (B) GFP 41-mer variants were analyzed for cleavage by SDS-PAGE and western blotting with anti-myc antibody as described in Fig 1 (C) Schematic of deletion and corresponding alanine extension variants. (D) ZMPSTE24-dependent cleavage of alanine extension variants was assessed by SDS-PAGE and western blotting with anti-myc antibody as described in Fig 2.

not required, and, more generally, that substituting alanines for the endogenous residues of prelamin A has only a modest impact on cleavage. Presumably the length requirement reflects the fact that folded GFP sterically impedes entry of the prelamin A C-terminus into the ZMPSTE24 chamber when it is too close to the cleavage site.

In parallel to the experiments above, and to independently determine if any specific residues in the region N-terminal to the cleavage site were necessary for substrate recognition and cleavage by ZMPSTE24, we also generated a set of constructs with multi-alanine replacements for residues R624-V629, G630-S636, and F637-V642, avoiding changing the residues TRSY immediately adjacent to the cleavage site (Fig 6A). Of these constructs, the 624–629 and 630–636 alanine substitutions showed considerable cleavage (~90% and 60% of WT, respectively (Fig 6B, compare lanes 2 and 3 to lane 1)). The reason that substitutions at residues 630–636 resulted in lower overall protein levels is unclear, but this phenomenon resembles what we observed for some of the deletion constructs, mentioned above (Fig 4 lanes 2–7). The alanine replacement of residues F637-V642 resulted in a diminished level of processing (just 43% that of wild-type (Fig 6B, lane 4)), which demonstrates that processing occurs, albeit at substantially reduced efficiency.

To determine if any specific residue in this region was particularly sensitive to substitution by alanine, we generated a series of single alanine replacements for residues F637-V642 (Fig 6C). However, no single alanine replacement resulted in a robust decrease in processing compared to WT (Fig 6D); all of the single substitutions are processed at >80% of the WT efficiency. Thus, additive or synergistic effects must account for the strong diminution in processing that we observed for the F637-V642 alanine substitutions (Fig 6D).

Overall, this analysis of the amino acids N-terminal to the ZMPSTE24 cleavage site in the GFP- 41-mer "minimal substrate" indicates that a tightly folded domain too close to the cleavage site can impede processing and that specific amino acids in this region are not required for cleavage, except those that impinge upon the cleavage site ($_{643}$TRSY^LLGN$_{650}$), since alanine substitutions between R624 and L642 have only a modest, if any, impact on cleavage.

## Discussion

### A surprising degree of flexibility for prelamin A processing by ZMPSTE24

The goal of this study was to determine what features of prelamin A, the sole known substrate of mammalian ZMPSTE24, are critical for its recognition and cleavage *in vivo*. Determining how ZMPSTE24 accesses its substrate and positions it for cleavage, as well as understanding the role of farnesyl are important for defining mechanistic features of this novel chambered protease and for revealing insight into progeroid diseases and possibly physiological aging. This information may also allow us to better predict other substrates of ZMPSTE24.

Our analysis focused on the C-terminal 41 residues of prelamin A (R624-M664), since we previously showed that a GFP-41-mer was a "minimal substrate" for efficient cleavage *in vivo* [47]. Here we tested whether there were specific residue or length requirements on either side of the cleavage site in this region of prelamin A. Taken together, we found a surprising degree of flexibility for what is allowable for ZMPSTE24 cleavage in the regions both C-terminal and N-terminal to the cleavage site. In general, most of the native residues of prelamin A between the cleavage site ($T_{643}$RSY^LLGN$_{650}$) and the farnesylated cysteine (C664) could be replaced by alanines with only modest detriment to cleavage, and expanding this region with additional alanines was also forgiving. Likewise, all of the residues N-terminal to the cleavage site could also be replaced with alanines with little impact on cleavage. Below we discuss our findings in the context of prior work. We also discuss how these results allow us to propose several possible working models for how ZMPSTE24 functions (Fig 7), and why farnesylated proteins other than prelamin A may not be substrates for this protease.

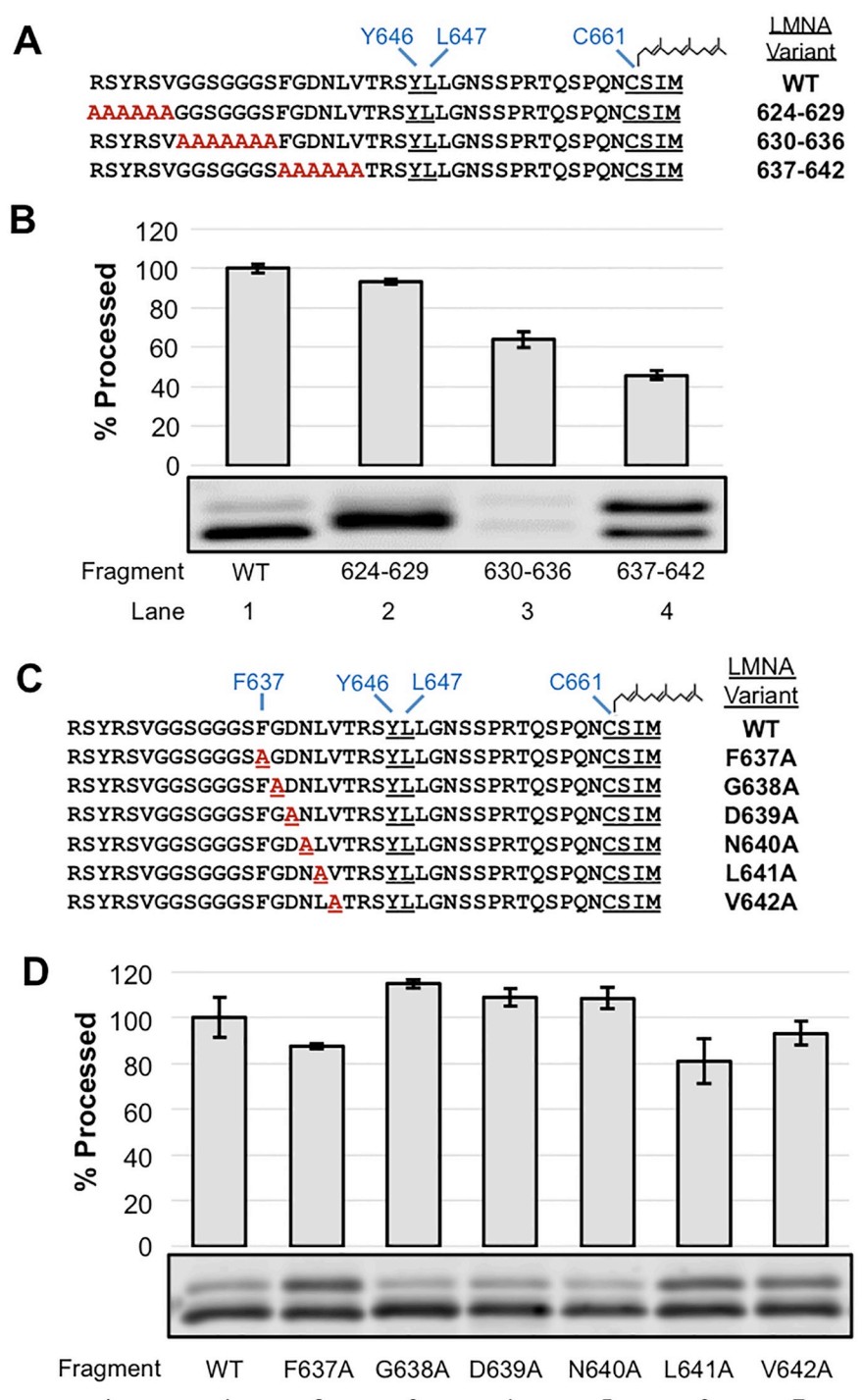

**Fig 6. Effect of alanine substitutions in the region N-terminal to the ZMPSTE24 cleavage site in the GFP-*LMNA*-41-mer substrate.** (A) GFP-41mer variants with multiple alanine substitutions upstream of the ZMPSTE24 cleavage site are shown. (B) Cleavage of GFP-41mer and the variants shown in (A) was assessed as described in Fig 2. Variation in 630-636A substrate expression level is not due to a difference in protein loading (S2 Fig) (C) Single alanine substitutions in the region between F637 and V642 are shown. (D) GFP-41mer variants were assayed as above.

### The composition and length of the region C-terminal to the cleavage site can vary considerably

It has been proposed that the C-terminus of prelamin A may need to be seated within the ZMPSTE24 chamber, binding to several residues in an elongated groove along the side of the chamber interior, with a hydrophobic patch at the bottom of the chamber binding to farnesyl (Fig 7A) [19]. However, we found that between the farnesylated cysteine and the cleavage site,

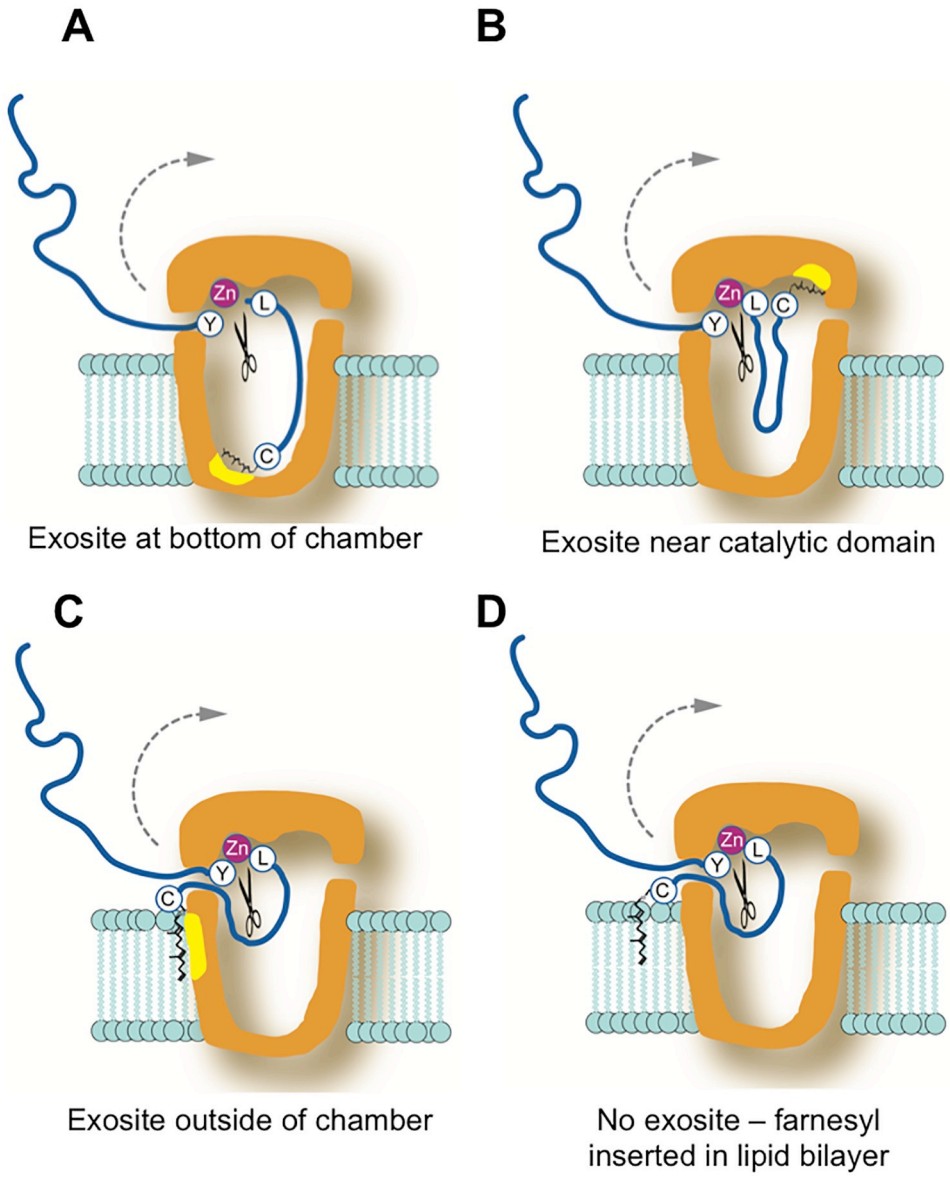

**Fig 7. Models for interaction between ZMPSTE24 and the C-terminal farnesylated portion of prelamin A.** In these models, the cleavage site between Y646 and L647 in prelamin A is shown positioned at the zinc metalloprotease catalytic domain in ZMPSTE24. The model of ZMPSTE24 is based on its known X-ray crystal structure [5]. Yellow patches denote possible exosites. The farnesylated cysteine (represented by C with a farnesyl group) may help orient prelamin A for cleavage by binding an exosite at the bottom of the chamber (A) or nearby the catalytic domain (B). Alternative models include farnesyl binding an exosite outside the ZMPSTE24 chamber (C) or farnesyl acting as a membrane-targeting signal and inserting into the lipid bilayer (D). The models shown here are discussed in more detail in the text.

step-wise alanine mutations that substituted up to 9 of the 14 native prelamin A residues did not dramatically impede cleavage (for instance, "9A" is cleaved at 60% the efficiency of WT, Fig 2). These data are in agreement with our previous result that prelamin A cleavage in HEK293 cells still occurred when we replaced the same 9 residues with an HA epitope (YPYDVPDYA) [47]. Importantly, these findings indicate that the nine amino acids immediately upstream of the farnesylated cysteine do not contain specific information for binding to a putative elongated exosite along the inner wall of the ZMPSTE24 chamber (Fig 7A). Only when the stretch of alanine substitutions encroached on the cleavage site of prelamin A (i.e. "11A", which replaces N650, and beyond, Fig 2) is processing dramatically decreased, which is unsurprising, since even a single N650A substitution near the cleavage site is known to impede prelamin A processing [24, 47].

Another model that has also been considered is that ZMPSTE24 acts as a molecular ruler, recognizing the farnesylated cysteine and then simply cleaving at an "allowable" cleavage site a particular distance away. However, we found here by elongating or shortening the C-terminal region of prelamin A, through adding alanines or deleting residues upstream of the farnesylated cysteine, that ZMPSTE24 is remarkably accommodating to increases or decreases in substrate length. We had previously observed that a 9 amino acid duplication of the native residues of prelamin A in this region diminished processing to some extent in mammalian cells, but some processing could still occur [47]. Here, we systematically increased the number of alanines to 19 in step-wise fashion and found that even after doubling the length of this region to 28 (by addition of 14 alanines), prelamin A processing still occurred (> 50% that of WT) (Fig 3). With additional alanines there was a notable and gradual decrease in processing, until cleavage is mostly abolished upon addition of 18 or 19 alanines (Fig 3). The reason for this cut-off in allowable length is not clear, since the chamber of ZMPSTE24 is large enough to accommodate even much longer C-terminal polypeptides [19]. Thus, the additional residues in the 18A and 19A constructs could impose structural constraints at the prelamin A C-terminus that interfere with cleavage. In any case, our analysis shows that while there is considerable flexibility in the allowable length of the substrate between the farnesylated cysteine and the cleavage site, there are nevertheless distinct limits.

Interestingly, we also found flexibility in this region in terms of shortening the prelamin A substrate. Deletion of the 5 of the 14 residues between the scissile bond and the farnesylated cysteine allowed considerable processing (>60%) (Fig 4), which is consistent with the alanine substitution experiments discussed above in which specific sequence information in this region was not essential for cleavage. However further deletions greatly diminished processing, although they did not entirely ablate it (Fig 4). Taken together, these results suggest that the precise length of the C-terminal tail is not critical for substrate processing, although excessive lengthening or shortening of this region can disrupt cleavage, and make the model shown in Fig 7A unlikely (see further discussion of this point below).

**Specific residues N-terminal to the ZMPSTE24 cleavage site of prelamin A are not critical for processing, but introduction of a folded domain close by can interfere with cleavage.** A similar picture of flexibility within limits and a lack of the need for specific residues, emerged from our analysis of cleavage requirements in the region N-terminal to the cleavage site of prelamin A. Our previous studies had shown that the last 41 residues of prelamin A fused to GFP (GFP-41mer) contains all of the information needed for ZMPSTE24 processing in mammalian cells, but that processing was diminished for a GFP-31-mer and a GFP-29-mer could not be cleaved at all [47]. Here we corroborated this result in the humanized yeast system, and went on to replace the deleted residues with alanines, creating "alternative 41-mers". In every case, cleavage was completely or significantly restored in the "alternative 41-mers" (Fig 5). Apparently, a folded domain such as GFP can impede entry into the ZMPSTE24 chamber

or in some other way prevent the prelamin A segment of the fusion from reaching the active site if GFP is too close to the farnesylated C-terminus. Full-length prelamin A contains an immunoglobulin-like (Ig) fold at residues 431–544 [25], However the Ig domain ends sufficiently far from the cleavage site (~100 amino acids away) that it apparently does not impact processing.

Notably, cleavage assays *in vitro* with synthetic farnesylated 20-mer and 29-mer prelamin A peptides as substrates and purified ZMPSTE24 demonstrated that the 29-mer and even the 20-mer can be efficiently and accurately cleaved *in vitro* [19, 57], echoing our result that no specific information N-terminal to the cleavage site is inherently required for ZMPSTE24 processing.

## How might ZMPSTE24 avoid farnesylated proteins in the cell other than prelamin A?

The dissection of prelamin A substrate requirements performed here and previously [47, 50] establishes some "guidelines" governing cleavage *in vivo*, including the need for a farnesylated cysteine with an allowable ZMPSTE24 cleavage site (minimally specified by a pair of hydrophobic residues, [24]) lying ~9–28 residues away from the farnesyl, the absence of a nearby folded domain, and surprisingly, few if any additional specific amino acids at particular positions.

With these minimal guidelines in mind, it is reasonable to consider why other CAAX-containing proteins (for instance lamin B, Ras proteins, or progerin), avoid processing by ZMPSTE24. The nuclear lamins B1 and B2 are farnesylated and have a potential di-hydrophobic cleavage site 14 residues away from their farnesylated cysteine, (comprised of leucine and phenylalanine). However, the presence of long stretches of negatively charged residues in both of the B-type lamins in this region may preclude entry into or proper positioning within the ZMPSTE24 chamber. Likewise, for Ras proteins, which are farnesylated, access to ZMPSTE24 may be precluded by nearby palmitoylation or the presence of a polybasic stretch in H-Ras and K-Ras, respectively [29]. Progerin is the disease-causing prelamin A deletion mutant associated with HGPS. In progerin, the CAAX box is present but 50 amino acids nearby are deleted, including the native ZMPSTE24 processing site, due to a splicing mutation [10, 11]. Progerin is farnesylated, and has a potential ZMPSTE24 cleavage site (two hydrophobic residues (valine and leucine) located 24 amino acids away from its farnesylated cysteine. Although this distance between the farnesylated cysteine and the putative cleavage site is on the longer end of what was allowable for prelamin A, the latter was still processed when the spacing was 28 residues (Fig 3B). However, progerin is not cleaved in mammalian cells [16], nor is it cleaved in our yeast system (S1 Fig). Thus, there may be a structure incompatible with entry into ZMPSTE24 or specific residues in progerin that actively prevent cleavage. Ultimately, additional studies are needed to understand whether interference is a major feature that prevents farnesylated proteins other than prelamin A from being ZMPSTE24 substrates. Indeed, for most CAAX proteins, cleavage of the region containing their farnesylated cysteine would interfere with their membrane localization and negate their biological activities [29].

Although blocking farnesylation of prelamin A either through mutagenesis or pharmacologically prevents cleavage by ZMPSTE24 *in vivo* [48–50], both farnesylated and non-farnesylated synthetic prelamin A peptides can be cleaved *in vitro*. However only farnesylated prelamin A peptides exclusively give rise to correctly processed cleavage products [19, 57], indicating that farnesylation is important for cleavage fidelity. Interestingly, although the farnesylated 15-mer cleavage product can remain associated with ZMPSTE24 *in vitro* [58], it has never been detected *in vivo*, suggesting it may be released into the lipid bilayer and rapidly

degraded. It is worthwhile noting that ZMPSTE24 and its yeast counterpart Ste24 have been genetically implicated in proteolytic roles in addition to cleavage of farnesylated prelamin A and **a**-factor, involving still undefined degradative roles in protein quality control and protein translocation and secretion [38, 39, 41, 43]. Furthermore *in vitro* studies have shown that non-farnesylated peptides unrelated to **a**-factor or prelamin A can also be cleaved by membrane preparations containing Ste24 [59], but little is known about how or whether this latter finding relates to Ste24's still ill-defined *in vivo* degradative functions in addition to its specific role in a-factor maturation.

## Models for prelamin A processing by ZMPSTE24 *in vivo*

Several models can be considered for how prelamin A is positioned within ZMPSTE24 for cleavage and the role of its farnesylated cysteine (Fig 7A–7D). While not mutually exclusive, the farnesyl moiety can serve different functions in these models. Farnesyl could bind an exo-site within the ZMPSTE24 chamber, helping to align the prelamin A substrate for cleavage. A hydrophobic patch consisting of 18 phenylalanines, leucines and isoleucines located near the ZMPSTE24 chamber floor has been suggested to serve in this role (Fig 7A) [19], although the function of this patch has yet to be explored by specific mutagenesis of the residues that comprise it. Our finding that deletion constructs are still cleaved makes such a distal exosite unlikely. However, an exosite hypothesized to be near the catalytic domain of ZMPSTE24 (Fig 7B) could accommodate our findings here that both deletions and extensions in the region are well-tolerated. Another possibility is that farnesyl embedded in the membrane could bind a region on the outer surface of the ZMPSTE24 chamber (Fig 7C), allowing the C-terminal portion of prelamin A to "back into" into the chamber to undergo cleavage. Alternatively, because it increases hydrophobicity, in combination with carboxymethylation, the role for farnesyl may simply be to allow the targeting of prelamin A to the ER/INM where ZMPSTE24 resides, bringing the substrate and enzyme in close proximity (Fig 7D), a requirement which can be overridden *in vitro* with exceedingly high concentrations of enzyme and substrate in a detergent-based environment. Further studies will be needed to determine which of these models is correct.

## Concluding remarks

Membrane-associated proteases like ZMPSTE24 play an important role in numerous biological processes and deciphering the rules for the substrate recognition of such proteases is an important, but often challenging, goal. For example, studies of the membrane protease presenilin that cleaves the Alzheimer's precursor protein have involved hundreds of researchers over the last two decades and, while much has been learned about substrate recognition and the choice of cleavage sites, many questions remain [60, 61]. For both proteases, presenilin and ZMPSTE24, early onset genetic aging disorders map to the genes that encode them, and defective function of both during normal physiological aging have been proposed to be detrimental to the health of aging individuals [18, 60, 61]. With the structure of ZMPSTE24 in hand and the information on its cleavage requirements from this and other studies at least partly established, we are poised to gain an even more detailed understanding of how this protease works.

## Supporting information

**S1 Fig. Progerin is not cleaved in the humanized yeast cleavage assay.** (A) Schematic of prelamin A and progerin substrates. Both have an N-terminal $His_{10}myc_3$-epitope tag for detection by western blotting. The ZMPSTE24 cleavage site is marked by an arrow in between Y646 and L647, which is missing in progerin due to a 50 amino acid deletion. (B) *ste24Δ* (SM4826) and *ste24Δ HA-ZMPSTE24* (SM6303) expressing prelamin A or progerin were analyzed by

SDS-PAGE and western blotting using anti-myc and anti-HA antibodies. Uncleaved (open arrowhead) and cleaved (black arrowhead) prelamin A forms are marked. No cleaved progerin was detected.
(TIFF)

**S2 Fig. Effect of alanine substitutions in the region N-terminal to the ZMPSTE24 cleavage site in the GFP-LMNA-41-mer substrate.** (A) GFP-41mer variants with multiple alanine substitutions upstream of the ZMPSTE24 cleavage site are shown. (B) Cleavage of GFP-41mer and the variants shown in (A) was assessed as described in Fig 2. Variation in 630-636A substrate expression level is not due to a difference in protein loading.
(TIFF)

**S1 Raw Images.**
(PDF)

## Acknowledgments

We thank members of the Carpenter Lab including Elisabeth Carpenter, Laiyin Nie and Ashley Pike, as well as members of the Michaelis lab for fruitful discussions.

## Author Contributions

**Conceptualization:** Susan Michaelis.

**Data curation:** Kaitlin M. Wood.

**Formal analysis:** Kaitlin M. Wood, Wenxin Xu.

**Funding acquisition:** Susan Michaelis.

**Investigation:** Kaitlin M. Wood, Eric D. Spear, Otto W. Mossberg, Kamsi O. Odinammadu, Wenxin Xu.

**Methodology:** Eric D. Spear.

**Project administration:** Susan Michaelis.

**Supervision:** Susan Michaelis.

**Writing – original draft:** Kaitlin M. Wood, Eric D. Spear, Susan Michaelis.

**Writing – review & editing:** Kaitlin M. Wood, Eric D. Spear, Susan Michaelis.

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
