## [Decision Letter · Decision Letter 0]

6 Oct 2020

PONE-D-20-28619

Defining Substrate Requirements for Cleavage of Farnesylated Prelamin A by the Integral Membrane Zinc Metalloprotease ZMPSTE24

PLOS ONE

Dear Dr. Michaelis,

Thank you for submitting your manuscript to PLOS ONE. After careful consideration, we feel that it has merit but does not fully meet PLOS ONE’s publication criteria as it currently stands. Therefore, we invite you to submit a revised version of the manuscript that addresses the points raised during the review process.

We look forward to receiving your revised manuscript.

Kind regards,

Albert Jeltsch

Academic Editor

PLOS ONE

Journal Requirements:

2.PLOS ONE now requires that authors provide the original uncropped and unadjusted images underlying all blot or gel results reported in a submission’s figures or Supporting Information files. This policy and the journal’s other requirements for blot/gel reporting and figure preparation are described in detail at https://journals.plos.org/plosone/s/figures#loc-blot-and-gel-reporting-requirements and https://journals.plos.org/plosone/s/figures#loc-preparing-figures-from-image-files. When you submit your revised manuscript, please ensure that your figures adhere fully to these guidelines and provide the original underlying images for all blot or gel data reported in your submission. See the following link for instructions on providing the original image data: https://journals.plos.org/plosone/s/figures#loc-original-images-for-blots-and-gels.

3.Thank you for stating the following financial disclosure:

 [This work was funded by a National Institute of Health (NIH) grant to SM (5R35GM127073).  Kaitlin Wood was funded in part by a Hay Fellowship from the Department of Cell Biology, Johns Hopkins School of Medicine. The funders had no role in study design, data collection and analysis, decision to publish, or preparation of the manuscript.].

We note that one or more of the authors is affiliated with the funding organization, indicating the funder may have had some role in the design, data collection, analysis or preparation of your manuscript for publication; in other words, the funder played an indirect role through the participation of the co-authors. If the funding organization did not play a role in the study design, data collection and analysis, decision to publish, or preparation of the manuscript and only provided financial support in the form of authors' salaries and/or research materials, please do the following:

Review your statements relating to the author contributions, and ensure you have specifically and accurately indicated the role(s) that these authors had in your study. These amendments should be made in the online form.

Confirm in your cover letter that you agree with the following statement, and we will change the online submission form on your behalf:

Reviewers' comments:

Reviewer's Responses to Questions

**Comments to the Author**

1. Is the manuscript technically sound, and do the data support the conclusions?

Reviewer #1: Yes

Reviewer #2: Yes

Reviewer #3: Yes

2. Has the statistical analysis been performed appropriately and rigorously? 

Reviewer #1: Yes

Reviewer #2: Yes

Reviewer #3: Yes

3. Have the authors made all data underlying the findings in their manuscript fully available?

Reviewer #1: Yes

Reviewer #2: Yes

Reviewer #3: Yes

4. Is the manuscript presented in an intelligible fashion and written in standard English?

Reviewer #1: Yes

Reviewer #2: Yes

Reviewer #3: Yes

5. Review Comments to the Author

Reviewer #1: The proteolytic processing of prelamin A by ZMPSTE24 is essential for the synthesis of lamin A, a key component of the nuclear lamina. Failure of this proteolytic step results in the accumulation of permanently farnesylated forms of prelamin A which cause the human genetic disease HGPS, and other progeroid disorders (e.g., RD). Defining the requirements for prelamin A processing are important, and clinically relevant. Using an established humanized yeast system, the authors define the substrate requirements for the terminal cleavage of farnesyl-prelamin A by ZMPSTE24. Based on systematic analysis of a region surrounding the cleavage site (spanning 38-amino acids), they define critical features (residues and length) for substrate recognition and cleavage. Unexpectedly, they show that the requirements are more flexible than expected. This is surprising considering that prelamin A is the only known substrate for ZMPSTE24 in mammalian cells. The manuscript is clearly written, the experimental system is appropriate, and the studies performed well. I had only a few comments.

1. After ZMPSTE24 cleaves prelamin A, what is the fate of the C-terminal fragment? Do the authors know if any of the C-terminal modifications affect the “exit” of the cleaved fragment from the ZMPSTE24 chamber? Could this indirectly affect the efficiency of prelamin A processing?

Minor comments:

1. PPrelamin A cleavage depends on the farnesylation of the CAAX motif. Although extremely unlikely, is it possible that some of the modifications affect prelamin A farnesylation (and affect prelamin A processing)?

2. The authors mention that low substrate expression in the yeast system does not appear to affect processing efficiency; however, is the opposite also true?

Reviewer #2: Notes for the Authors:

In this study, the authors use a humanized yeast assay, with stable expression of human ZMPSTE24, while delivering epitope-tagged minimal C-terminal Lamin A constructs, to evaluate the substrate requirements for this protease. Despite the fact that ZMPSTE24 is only known to cleave a single substrate in vivo, the authors observe remarkable flexibility in the actual substrate sequence specificity outside of the known required P1’ and P2’ sites. In addition, they observe clear restrictions on substrate length that argues against a cleavage model with an exosite for the farnesylated cysteine at the protease chamber floor. Overall, the authors present a concise and well-written story that is supported by the data presented. While the story would feel more complete if there were data to specifically advocate for one of the several proposed cleavage models, it is recognized that doing so would likely require an extensive investigation best served as its own stand-alone manuscript.

Several issues are raised for consideration:

- As the authors acknowledge, several of the LaminA constructs are poorly expressed (LMNA 630-636, and LMNA ∆2 through ∆6), though the authors are still able to quantify the relative cleavage. There is a possibility, especially since the WT construct is not 100% cleaved, that the amount of substrate delivered in vivo in the yeast system is saturating, which could cause under-reporting of actual cleavage that happens in a natively expressed system. This would also make the reporting of the relative cleavage of poorly expressed constructs appear better. Do the authors have data to suggest for or against this? This might be answered by delivering the substrate under a less robust promoter, but also might have been tested in another system by the authors that is not presented here.

- Given the data provided in the paper, the reasons that ZMPSTE24 does not cleave progerin is unknown, since it has a putative minimal cleavage sequence (valine at P1’ and leucine at P2’) 24 amino acids upstream from the farnesylated cysteine. The authors appropriately acknowledge this in the discussion (page 19). The study would be significantly strengthened by replacing the 8 AA spacer between the required P2’ residue and the progerin splice site (i.e. the GNSSPRTQ sequence in the WT construct) with the corresponding possible progerin spacers (C-terminal: ASGSGAQV; or N-terminal: CGTCGQPA). This could provide supportive evidence as to whether specific residues actively prevent cleavage (in which case, one or both of these progerin spacers should preclude cleavage, irrespective of the shorter length of this spacer).

- A control to show that the actual progerin sequence does not get cleaved when it replaces the minimal 41-AA WT Lamin A sequence would be very helpful to confirm the relevance of the system to the disease process at hand.

Minor Suggestions:

- The title of Fig. 1 (ZMPSTE24-dependent prelamin A cleavage can be recapitulated using a humanized yeast assay) implies a conclusion from data. A title that reflects the schematics shown would be more appropriate.

- Fig. 2 shows a “representative” immunoblot which itself is a composite image from separate gels. For publication purposes, it would be best to have a blot with all experimental conditions on the same gel.

- The hexokinase loading controls for the gels are not provided but should be peer reviewed for completeness.

- The legends of Figs. 3-5 reference the legend of Fig. 1, but likely mean to reference the legend of Fig. 2 (to illustrate 3 separate independent experiments, etc).

- Can the ZMPSTE24 chamber permit formation of secondary structure? It seems possible that the +ala constructs (31+ala, 29+ala, 27+ala) from Figs. 5C&D may allow the spontaneous formation of an alpha helix, and a flexible amino acid linker (some combination of glycines and serines) could be a nice comparison to probe this possibility.

- For future studies, it is intriguing that Ste24 (the yeast protease) is able to perform -AAX removal. Is the human ZMPSTE24 able to do this as well? If yes, what does that say about access to the ZMPSTE24 active site? If not, can any differences between the yeast and human versions inform why?

Reviewer #3: The manuscript “Defining Substrate Requirements for Cleavage of Farnesylated Prelamin A by the Integral Membrane Zinc Metalloprotease ZMPSTE24” by Wood et al. describes a systematic study of the critical features of prelamin A as the substrate of mammalian metalloprotease, ZMPSTE24, using a well-established humanized yeast platform. This is a continuation and expansion of their previous work: Methods. 2019 Mar 15; 157: 47–55. By inserting, replacing or truncating residues around proteolytic sites, the authors confirmed a high degree of flexibility for what is allowable for ZMPATE24 cleavage in the regions. Based on what they have found in this and their previous work, the authors came up with several possible working models for ZMPSTE24 functions. Overall, the methodology worked out smoothly and properly. The paper is well written, the results support the conclusion and the discussion is thorough and intriguing. I have the following comments, most of which are minor and easy to address.

1. For the cleavage models in Fig 7A and 7B where the Farnesylated tail of Prelamin A binds somewhere in the chamber interior that is required for cleavage, did the authors think about how the release of the cleaved -LLGC(Farnesyl) regulates the proteolytic efficiency of ZMPSTE24. The binding affinity of the Farnesylated tail to the interior chamber matters because only if the proteolytic C-terminal is released from the binding site in chamber, can the ZMPSTE24 proceed to another catalytic turn-over. Please discuss possible product inhibition issues. These can be measured by progress curve analysis, but may be outside the scope of this work.

2. The crystal structure is available for ZMPSTE24. Is it possible to calculate the size of the chamber and estimate how many amino acids can be fit into it. This number can help to rule out some models in Fig 7.

3. The authors should be more cautious when claiming that amino acid residues around the cleavage site are not important for ZMPSTE24 proteolytic processing. The authors only explored alanine for either mutations or insertions.

4. Can the authors comment on the possibility that other post-translational modifications (PTMs) on the 41-amino acid segment may also regulate the proteolytic processing. For example it is known that phosphorylation can regulate cleavage by caspases. In addition, since the C-terminal tail is flexible and does not have any folded structure, is it possible that there are other protease (s) that can cleave it in vivo?

5. Much of the quantitative data shown in this manuscript is based upon western blot. Please provide the method details for WB quantification in Material and Method section.

6. Minor issues:

o There are two affiliations (1 & 2) listed on the title page but none of the authors have “2” as superscript.

o Not sure if the figures I can download are the final ones, but they are of very low resolution.

o Fig 1A legend “The lamin A precursor prelamin A is 664 amino acids in length, and after CAAX processing is farnesylated and carboxymethylated C-terminus at cysteine 661, as shown.” This sentence is awkward; please rephrase it.

6. PLOS authors have the option to publish the peer review history of their article (what does this mean?). If published, this will include your full peer review and any attached files.

Reviewer #1: No

Reviewer #2: **Yes: **Charles S. Craik, PhD

Reviewer #3: No

---

## [Author Response · Author response to Decision Letter 0]

20 Nov 2020

Please see below our responses underneath each of the Reviewers' comments. We thank the Reviewers for useful suggestions.

Reviewer #1: 

The proteolytic processing of prelamin A by ZMPSTE24 is essential for the synthesis of lamin A, a key component of the nuclear lamina. Failure of this proteolytic step results in the accumulation of permanently farnesylated forms of prelamin A which cause the human genetic disease HGPS, and other progeroid disorders (e.g., RD). Defining the requirements for prelamin A processing are important, and clinically relevant. Using an established humanized yeast system, the authors define the substrate requirements for the terminal cleavage of farnesyl-prelamin A by ZMPSTE24. Based on systematic analysis of a region surrounding the cleavage site (spanning 38-amino acids), they define critical features (residues and length) for substrate recognition and cleavage. Unexpectedly, they show that the requirements are more flexible than expected. This is surprising considering that prelamin A is the only known substrate for ZMPSTE24 in mammalian cells. The manuscript is clearly written, the experimental system is appropriate, and the studies performed well. I had only a few comments.

1. After ZMPSTE24 cleaves prelamin A, what is the fate of the C-terminal fragment? Do the authors know if any of the C-terminal modifications affect the “exit” of the cleaved fragment from the ZMPSTE24 chamber? Could this indirectly affect the efficiency of prelamin A processing

 We appreciate the Reviewer’s thoughtful comment (and have now added a sentence to the text to clarify the point the reviewer raises, as noted below). The answers to the Reviewer’s two questions posed are likely NO and NO for the following reasons: Despite intense experimental attempts by our lab and many others, the cleaved C-terminal fragment has never been detected in vivo even though high quality antibodies raised to this fragment easily detect it on prelamin A prior to cleavage. This has led to the general consensus that this C-terminal farnesylated fragment is rapidly degraded after cleavage in vivo. In contrast, as reported by Mehmood et al (PMID: 27874871), in an in vitro cleavage reaction using purified ZMPSTE24 in detergent micelles and a synthetic farnesylated peptide substrate, the C-terminal fragment can stay associated with ZMPSTE24 post-cleavage, as detected by high resolution mass spectrometry. However, the authors of that report pointed out that absence of a lipid bilayer in that in vitro study may have allowed them to capture a normally transient intermediate that is quickly resolved in vivo through release of the fragment into the ER membrane bilayer or lumen, followed by degradation. We have now added a sentence to page 20 that notes that the C-terminal fragment appears to be rapidly degraded in vivo.

.

Minor comments:

1. PPrelamin A cleavage depends on the farnesylation of the CAAX motif. Although extremely unlikely, is it possible that some of the modifications affect prelamin A farnesylation (and affect prelamin A processing)?

 As the Reviewer states that while it is possible, it is highly unlikely that any of our mutations affect cleavage indirectly, by altering farnesylation (and we now added a sentence and references to this point, as noted below). Numerous studies in the farnesylation field over decades indicate that it is solely the specific residues within the CAAX motif that determine efficiency and type of prenylation. In addition, farnesylation generally causes altered migration of a protein in SDS-PAGE gels, and we did not detect this for the mutants we analyzed. Nevertheless, the Reviewer’s point is well taken and we have now added a sentence on page 9 mentioning that a potential caveat is that some mutations at a distance from the CAAX motif could conceivably influence farnesylation although this is highly unlikely and we also added 3 references supporting the accepted wisdom that CAAX motifs are the sole determinant for dictating farnesylation status (PMIDs 25402849, 30257935, and 26790532).

2. The authors mention that low substrate expression in the yeast system does not appear to affect processing efficiency; however, is the opposite also true?

 The level of ZMPSTE24 determines the steady-state level of cleavage (Spear et al, 2019; PMID 30625386). We have chosen in this study to use a level of ZMPSTE24 expression (2 chromosomal copies per cell) that maximizes our ability to discriminate degrees of processing through a wide range of substrate levels. We also address this point below for Reviewer 2’s first comment. 

Reviewer #2:

In this study, the authors use a humanized yeast assay, with stable expression of human ZMPSTE24, while delivering epitope-tagged minimal C-terminal Lamin A constructs, to evaluate the substrate requirements for this protease. Despite the fact that ZMPSTE24 is only known to cleave a single substrate in vivo, the authors observe remarkable flexibility in the actual substrate sequence specificity outside of the known required P1’ and P2’ sites. In addition, they observe clear restrictions on substrate length that argues against a cleavage model with an exosite for the farnesylated cysteine at the protease chamber floor. Overall, the authors present a concise and well-written story that is supported by the data presented. While the story would feel more complete if there were data to specifically advocate for one of the several proposed cleavage models, it is recognized that doing so would likely require an extensive investigation best served as its own stand-alone manuscript.

Several issues are raised for consideration:

- As the authors acknowledge, several of the LaminA constructs are poorly expressed (LMNA 630-636, and LMNA ∆2 through ∆6), though the authors are still able to quantify the relative cleavage. There is a possibility, especially since the WT construct is not 100% cleaved, that the amount of substrate delivered in vivo in the yeast system is saturating, which could cause under-reporting of actual cleavage that happens in a natively expressed system. This would also make the reporting of the relative cleavage of poorly expressed constructs appear better. Do the authors have data to suggest for or against this? This might be answered by delivering the substrate under a less robust promoter, but also might have been tested in another system by the authors that is not presented here.

 As noted above in answer to Reviewer 1, comment 2, we have shown previously that expressing different levels of ZMPSTE24 enzyme can tune levels of cleaved/uncleaved prelamin A (Spear et al, 2019; PMID 30625386), and we chose to use here a level of ZMPSTE24 that maximizes our ability to discriminate degrees of processing through a wide range of substrate levels. 

- Given the data provided in the paper, the reasons that ZMPSTE24 does not cleave progerin is unknown, since it has a putative minimal cleavage sequence (valine at P1’ and leucine at P2’) 24 amino acids upstream from the farnesylated cysteine. The authors appropriately acknowledge this in the discussion (page 19). The study would be significantly strengthened by replacing the 8 AA spacer between the required P2’ residue and the progerin splice site (i.e. the GNSSPRTQ sequence in the WT construct) with the corresponding possible progerin spacers (C-terminal: ASGSGAQV; or N-terminal: CGTCGQPA). This could provide supportive evidence as to whether specific residues actively prevent cleavage (in which case, one or both of these progerin spacers should preclude cleavage, irrespective of the shorter length of this spacer).

- A control to show that the actual progerin sequence does not get cleaved when it replaces the minimal 41-AA WT Lamin A sequence would be very helpful to confirm the relevance of the system to the disease process at hand.

 We thank the Reviewer for this comment and we have now added as supplemental data our negative result showing that progerin is not cleaved in this system (S1 Fig). We appreciate the Reviewer’s suggestions for trying to pinpoint the block in progerin processing by substituting segments of progerin into prelamin A. However, our ongoing experiments suggest the answer may be complicated. Namely, necessary sequences may be missing in progerin and in addition some progerin sequences may be inhibitory to processing. This is an ongoing separate body of work in the lab. Because the present study is focused on prelamin A and progerin is only mentioned as a minor point, we feel that a detailed analysis of why progerin does not get processed is outside the scope of the paper (and is accessory to the point of this study which is to define the allowable features that make prelamin A a ZMPSTE24 substrate).

Minor Suggestions:

- The title of Fig. 1 (ZMPSTE24-dependent prelamin A cleavage can be recapitulated using a humanized yeast assay) implies a conclusion from data. A title that reflects the schematics shown would be more appropriate. 

 We appreciate this suggestion and have now changed the title of Fig.1 on page 3.

- Fig. 2 shows a “representative” immunoblot which itself is a composite image from separate gels. For publication purposes, it would be best to have a blot with all experimental conditions on the same gel.

 We have clearly marked the lanes that were used to make the composite figure for Fig.2. As per PLoS One’s requirements for publication, we have also included all gels/blots used to compile the figures (S1_raw_images), including those that have all the variants loaded on a single gel for Fig 2. Because the latter gels happen not to be as “photogenic” as the one shown (and because a slightly different type of gels were used for those (commercial) versus the others in Figs 2-6 (home-made), we have opted to keep the composite version of Fig.2, for consistency with the other figures in this study.

- The hexokinase loading controls for the gels are not provided but should be peer reviewed for completeness.

 As stated above, we have now included all relevant blots used to compose figures, including the hexokinase loading controls for all in the file “S1_raw_images”. For simplification, and in accordance with the Reviewer’s request, we have also now added a supplemental figure (Supplemental Fig.S2) to show equal loading specifically for Fig. 6B.

- The legends of Figs. 3-5 reference the legend of Fig. 1, but likely mean to reference the legend of Fig. 2 (to illustrate 3 separate independent experiments, etc).

 We appreciate that the Reviewer caught this typo. We have now made the correction in the figure legends of Figs 3-6.

- Can the ZMPSTE24 chamber permit formation of secondary structure? It seems possible that the +ala constructs (31+ala, 29+ala, 27+ala) from Figs. 5C&D may allow the spontaneous formation of an alpha helix, and a flexible amino acid linker (some combination of glycines and serines) could be a nice comparison to probe this possibility.

 We agree this series of experiments could be of interest, especially since there is a flexible linker in the wild-type sequence. However, because we found in Fig 5 that replacing this sequence with alanines revived processing to a great extent, we feel that replacing these with a flexible linker, even if it improves cleavage even a bit more, may provide only incrementally more information. Thus we feel that generating these constructs seems outside the scope of the current study.

- For future studies, it is intriguing that Ste24 (the yeast protease) is able to perform -AAX removal. Is the human ZMPSTE24 able to do this as well? If yes, what does that say about access to the ZMPSTE24 active site? If not, can any differences between the yeast and human versions inform why?

 In answer to the first question, yes, human ZMPSTE24 and yeast Ste24 are remarkably similar in structure, and seem to have the same specificities for both –AAX removal and upstream (SY^LL) cleavage for yeast a-factor. Indeed, it was previously shown that human ZMPSTE24 can functionally replace yeast Ste24 in the formation of mature yeast a-factor, which has a CAAX sequence CVIA (Leung et al, 2001; PMID: 11399759). However, further studies by us and others have found that although ZMPSTE24/Ste24 can properly cleave the CVIA sequence of a-factor, it cannot cleave some other CAAX motifs with good efficiency and fidelity. Most relevantly, the native prelamin A CAAX sequence CSIM cannot be properly cleaved by ZMPSTE24 (Nie et al, bioRxiv; https://doi.org/10.1101/2020.05.13.093849). Long story short, it appears that for prelamin A, the other known protease RCE1 is the relevant CAAX processing enzyme, while ZMPSTE24 functions solely as the prelamin A SY^LL protease. Thus, the models shown in Fig. 7 only consider the access step for SY^LL cleavage.

Reviewer #3: 

The manuscript “Defining Substrate Requirements for Cleavage of Farnesylated Prelamin A by the Integral Membrane Zinc Metalloprotease ZMPSTE24” by Wood et al. describes a systematic study of the critical features of prelamin A as the substrate of mammalian metalloprotease, ZMPSTE24, using a well-established humanized yeast platform. This is a continuation and expansion of their previous work: Methods. 2019 Mar 15; 157: 47–55. By inserting, replacing or truncating residues around proteolytic sites, the authors confirmed a high degree of flexibility for what is allowable for ZMPATE24 cleavage in the regions. Based on what they have found in this and their previous work, the authors came up with several possible working models for ZMPSTE24 functions. Overall, the methodology worked out smoothly and properly. The paper is well written, the results support the conclusion and the discussion is thorough and intriguing. I have the following comments, most of which are minor and easy to address.

1. For the cleavage models in Fig 7A and 7B where the Farnesylated tail of Prelamin A binds somewhere in the chamber interior that is required for cleavage, did the authors think about how the release of the cleaved -LLGC(Farnesyl) regulates the proteolytic efficiency of ZMPSTE24. The binding affinity of the Farnesylated tail to the interior chamber matters because only if the proteolytic C-terminal is released from the binding site in chamber, can the ZMPSTE24 proceed to another catalytic turn-over. Please discuss possible product inhibition issues. These can be measured by progress curve analysis, but may be outside the scope of this work.

 We appreciate this comment about the fate of the cleaved farnesylated tail, which is also similar to that made by Reviewer 1, comment 1. As noted in our response to Reviewer 1, we have now added a sentence on page 20 discussing that fact that the cleaved 15-mer appears to be rapidly degraded in vivo. Thus, product inhibition in vivo as suggested by Reviewer 3 is unlikely. Regarding progress curve analysis, while at some point it would be good to compare mutants in vivo and in vitro by such analysis, we agree this in vitro analysis falls outside the scope of the current in vivo work.

2. The crystal structure is available for ZMPSTE24. Is it possible to calculate the size of the chamber and estimate how many amino acids can be fit into it. This number can help to rule out some models in Fig 7. 

 We also appreciate this comment. The size of the chamber of ZMPSTE24 and its homolog yeast Ste24 is surprisingly large -12000 to 14000 cubic angstroms, large enough to fit a 10-kD protein or 450 water molecules (Quigley et al; PMID: 23539603 and Pryor et al; PMDI: 23539602), so this does not rule out any of the models in Fig 7. The models shown in Fig. 7 are drawn “to scale”, based on the actual structure determined by crystallography, modeling the prelamin A C-terminus into that structure. We have now added a sentence clarifying these points in the Fig 7 legend on page 15. We also added a comment about the chamber size on page 4.

3. The authors should be more cautious when claiming that amino acid residues around the cleavage site are not important for ZMPSTE24 proteolytic processing. The authors only explored alanine for either mutations or insertions.

 We are unsure of exactly where the Reviewer felt that we overstated the case that amino acid residues around the cleavage site are not important for processing. We tried throughout, rather than claiming that certain residues are not important for cleavage to emphasize that there is considerable flexibility in terms of specific requirements. 

4. Can the authors comment on the possibility that other post-translational modifications (PTMs) on the 41-amino acid segment may also regulate the proteolytic processing. For example it is known that phosphorylation can regulate cleavage by caspases. In addition, since the C-terminal tail is flexible and does not have any folded structure, is it possible that there are other protease (s) that can cleave it in vivo?

 The Reviewer raises an interesting point that post translational modifications (PTMs) (aside from the CAAX modifications of farnesylation and methylation) could potentially influence prelamin A processing. However solid evidence for the presence of PTMs in the C-terminal region of prelamin A that influence processing is lacking. Some studies have detected O-GlycNac or phosphorylation modifications in this region by mass spectrometry (https://pubmed.ncbi.nlm.nih.gov/23475188/ and https://www.ncbi.nlm.nih.gov/pmc/articles/PMC5981268/). However, whether these modifications are present on a significant percentage of molecules is not at all clear, and no evidence suggests they may have a role in processing. Because PTMS in the C-terminal region of prelamin A (aside from farnesylation and methylation, of course!), are not well-established, we did not feel it was necessary to discuss this hypothetical possibility 

 Regarding the Reviewer’s second question of whether there are other proteases aside from ZMPSTE24 that can cleave prelamin A, the answer is NO, since in mice or cells lacking ZMPSTE24 prelamin A cannot be cleaved. Only farnesylated prelamin A is observed.

5. Much of the quantitative data shown in this manuscript is based upon western blot. Please provide the method details for WB quantification in Material and Method section.

 We appreciate this suggestion and have now added Methods details on page 8.

6. Minor issues:

o There are two affiliations (1 & 2) listed on the title page but none of the authors have “2” as superscript.

 We have now corrected this, adding the “2” to one of the authors. 

o Not sure if the figures I can download are the final ones, but they are of very low resolution.

 We have ensured that we uploaded high resolution figures on this resubmission.

o Fig 1A legend “The lamin A precursor prelamin A is 664 amino acids in length, and after CAAX processing is farnesylated and carboxymethylated C-terminus at cysteine 661, as shown.” This sentence is awkward; please rephrase it.

 We appreciate the Reviewer pointing out our awkward phrasing. We have now corrected this in the Fig 1A legend on pages 3-4.

---

## [Editor Report · Decision Letter 1]

24 Nov 2020

Defining Substrate Requirements for Cleavage of Farnesylated Prelamin A by the Integral Membrane Zinc Metalloprotease ZMPSTE24

PONE-D-20-28619R1

Dear Dr. Michaelis,

We’re pleased to inform you that your manuscript has been judged scientifically suitable for publication and will be formally accepted for publication once it meets all outstanding technical requirements.

Kind regards,

Albert Jeltsch

Academic Editor

PLOS ONE
---

## [Editor Report · Acceptance letter]

2 Dec 2020

PONE-D-20-28619R1 

Defining Substrate Requirements for Cleavage of Farnesylated Prelamin A by the Integral Membrane Zinc Metalloprotease ZMPSTE24 

Dear Dr. Michaelis:

I'm pleased to inform you that your manuscript has been deemed suitable for publication in PLOS ONE. Congratulations! Your manuscript is now with our production department. 

Kind regards, 

on behalf of

Prof. Dr. Albert Jeltsch 

Academic Editor

PLOS ONE